# Bacterial Diversity Correlates with Overall Survival in Cancers of the Head and Neck, Liver, and Stomach

**DOI:** 10.3390/molecules26185659

**Published:** 2021-09-17

**Authors:** Rebecca M. Rodriguez, Mark Menor, Brenda Y. Hernandez, Youping Deng, Vedbar S. Khadka

**Affiliations:** 1Bioinformatics Core, Department of Quantitative Health Sciences, John A. Burns School of Medicine, University of Hawaii Mānoa, Honolulu, HI 96813, USA; rebecca.rodriguez2@nih.gov (R.M.R.); mmenor@hawaii.edu (M.M.); 2Population Sciences in the Pacific Program-Cancer Epidemiology, University of Hawaii Cancer Center, Honolulu, HI 96813, USA; brenda@cc.hawaii.edu; 3National Institute of Diabetes and Digestive and Kidney Diseases, NIH, Bethesda, MD 20892, USA

**Keywords:** microbial diversity, tumor microenvironment, infection-associated cancers

## Abstract

One in five cancers is attributed to infectious agents, and the extent of the impact on the initiation, progression, and disease outcomes may be underestimated. Infection-associated cancers are commonly attributed to viral, and to a lesser extent, parasitic and bacterial etiologies. There is growing evidence that microbial community variation rather than a single agent can influence cancer development, progression, response to therapy, and outcome. We evaluated microbial sequences from a subset of infection-associated cancers—namely, head and neck squamous cell carcinoma (HNSC), liver hepatocellular carcinoma (LIHC), and stomach adenocarcinoma (STAD) from The Cancer Genome Atlas (TCGA). A total of 470 paired tumor and adjacent normal samples were analyzed. In STAD, concurrent presence of EBV and *Selemonas sputigena* with a high diversity index were associated with poorer survival (HR: 2.23, 95% CI 1.26–3.94, *p* = 0.006 and HR: 2.31, 95% CI 1.1–4.9, *p* = 0.03, respectively). In LIHC, lower microbial diversity was associated with poorer overall survival (HR: 2.57, 95% CI: 1.2, 5.5, *p* = 0.14). Bacterial within-sample diversity correlates with overall survival in infection-associated cancers in a subset of TCGA cohorts.

## 1. Introduction

Microbiological infections account for up to 20% of the total global cancer burden, one of the leading causes of morbidity and mortality worldwide [1,2]. Despite overall declining mortality rates in the United States, cancer remains the leading cause of death in the population including Asian Americans, Native Hawaiians, and persons of Hispanic origin [3,4]. Racial disparities persist in infection-associated cancers including liver, gastric, and head and neck [5]. Besides lung and breast cancers, liver, gastric, and head and neck cancers are in the top 10 causes of cancer-related deaths worldwide with marked racial differences. Underlying causes for racial disparities are multifactorial and not well understood [6,7,8]. Recent studies suggest a potential role of microbial composition in race-related disparities [9]. While much effort has gone into the characterization of the gut and oral microbiota, compositional differences of tumor tissue are less explored.

Identification of tissue-associated microbial differences is challenging and computationally intensive. There is growing evidence suggesting microbial communities have a dual modulating effect in cancer pathogenesis [10,11]. Viral-bacterial co-occurrence has been identified to modulate tumor aggressiveness [12]. Based on epidemiological and geographic correlations, it is suggested that viral agents may interact with specific bacteria, resulting in more aggressive tumors and poorer outcomes. For instance, it is recognized that Epstein–Barr virus (EBV) (also called HHV-4) infected stomach tumors are molecularly distinct, while its interaction with *Helicobacter pylori* remains unconfirmed. In liver hepatocellular carcinoma co-infection with hepatitis B virus (HBV) and hepatitis C virus (HCV) and their interaction between proteins can also lead to more aggressive tumors [13,14].

New evidence has hinted at the association of gut microbial dysbiosis with cancer clinical outcomes, even potentially influencing racial-related differences [15,16]. Gopalakrishnan et al. determined that a highly diverse gut microbiome can provide improved antitumor response, while low diversity with high abundance of unfavorable bacteria such as Bacteroidales may result in weakened antitumor presentation capacity [16]. We wanted to evaluate if similar patterns existed when examining the microbial composition within the tumor microenvironment and their influence in clinical outcomes and survival. To our knowledge, no studies have examined microbial within-sample diversity derived directly from the paired human tumor microenvironment and its relation to overall survival and potential impact on racial differences in HNSC, LIHC, and STAD cohorts. We aimed to study differences in tumor and normal tissue microbiome diversity and determine if this has any relationship with overall survival among infection-associated cancers.

## 2. Results

### 2.1. Population Characteristics and Microbial Diversity Profiles 

A total of 470 paired tumor and adjacent normal samples encompassing 235 cases from HNSC, LIHC, and STAD were examined. Of the 470 samples, 11% (1 from HNSC, 28 samples from LIHC and 23 from STAD) had no detectable bacteria presence, 7% of samples (7 from HNSC, 15 from LIHC, and 11 from STAD) had single bacteria presence. In both cases, diversity indices were represented as zero. All cases with at least one bacteria species present in either tumor or normal were considered for analyses as described in Rodriguez et al. [17]. Viral presence (HBV, HHV-4, or HPV) was detected in 31% of samples without bacterial presence. The sample population characteristics for HNSC, LIHC, and STAD cohorts and microbial diversity profiles stratified by sample type are summarized in Table 1 and Table 2, respectively.

### 2.2. Microbial Profiles Differed across Institutions

Potential contamination and laboratory artifacts may lead to microbial differences. In previous studies [18] comparing tumor and matched blood specimens, it was reported that microbial signatures including virus, bacteria, and other species related ethnic differences were in fact due to specific stamp signatures corresponding to the institutions where specimens originated or processed. Tae et al. concluded that these signatures could correspond to contamination or laboratory artifacts [18]. Based on this report, we evaluated whether the race-associated microbial differences were due to the institution from which they originated. Because of our strictly paired analyses, if microbial signatures are potentially contamination, we would expect the same rate of relative abundance patterns in tumor and adjacent normal paired samples. We compared relative abundance of specific taxa found to be prevalent and differentially present within the sample population across submitter site institutions and between samples from different racial groups originating from the same institution.

All HNSC specimens in our study originated and were processed at the same institution and no comparisons were performed. It is presumed that observed variation is attributed to racial and clinical presentation differences. LIHC samples originated from a diverse population enrolled at different institutions and processed at a single institution. We found no overall differences across submitter sites in pairwise comparison using Wilcox test with Bonferroni correction for multiple testing. In analyses of variance adjusting for race and sample type, some species were identified as significantly associated with submitter site, *Ralstonia pickettii* (F(10, 147) = 24.4, *p* < 0), *Klebsiella pneumoniae* (F(10, 147) = 1.97, *p* = 0.04), *Rhodococcus erythropolis* (F(10, 147) = 2.73, *p* = 0.004), and *Bradyrhizobium japonicum* (F(10, 147) = 2.23, *p* = 0.02). *Ralstonia pickettii* and *Rhodococcus erythropolis* associations were dependent on sample type, while *Escherichia coli*, the most abundant species detected in LIHC was associated with submitter site in linear regression models. A two-way ANOVA on relative abundance by submitter site and race revealed a statistically significant interaction between submitter site and race for *Escherichia coli* (F(7, 141) = 7.5, *p* = 0). A Tukey’s HSD post hoc test was carried out, which showed that the relative abundance of *Escherichia coli* differed significantly across institutions with a mean difference of 380.8, BH adjusted *p* = 0.04. This difference was driven by the White population originating from a single site.

STAD subsets of samples originated from different institutions were processed at the same institution but different from those in LIHC. However, racial minority groups in STAD were uniquely recruited from specific institutions. We presumed that institutions with only one racial group would be inherently different when examining population differences between institutions. Therefore, we compared the relative abundance of top-taxa within the background (White) population across institutions, and across the racial groups within the same institution. We found no evidence to suggest an interaction between the submitter site and racial groups on species relative abundances. When comparing White and non-White from the same institution, distinct patterns emerged. Despite, bacterial relative abundance differences between originating institutions, in pairwise comparison with Bonferroni correction, were not statistically different. We conclude that observed relative abundance differences between tumor and paired adjacent normal across racial groups in STAD are true observations not due to artifacts from the originating sites. Similar to prior gut microbiota studies, we find that relative abundances of certain taxa are associated with racial groups.

### 2.3. Microbial Relative Abundance Differed by Race

Microbial composition and relative abundance have been associated with race in healthy and cancer patients [9,15]. We compared microbial relative abundance stratifying by race at the genus level. Average relative abundance based on top 20 taxa per sample type illustrates some clear differences between tumor and adjacent normal in each racial group (Figure 1). Figure 1A shows the genus level average relative abundance differences across racial groups within HNSC, where *Sphingomonas* genus was present across all racial groups with the smallest proportion among Black or African American tissue samples. Tumor tissue samples from HNSC White and non-reported race groups appear to have a more diverse tumor microenvironment compared to tumor tissue from Asian and Black or African American cases. These groups also had fewer species compared to their own paired adjacent normal tissue and to White and non-reported racial groups.

Among the three cohorts, LIHC tissue samples had the lowest number of taxa identified at the genus level, with most reads mapped to *Escherichia coli* (Figure 1B), while STAD appeared to have a more diverse tumor microenvironment, especially among the Asian group (Figure 1C). Relative abundance patterns were similar across all racial groups, with paired tissue samples within the LIHC cohort with Black or African American group having a greater abundance of *Pseudomonas* and *Ralstonia* species. LIHC shared species in similar patterns with the HNSC cohort across Black or African American and White tissue samples, such as the presence of *Variovorax* which is not present in STAD.

### 2.4. Tumor Microbial Diversity Differs by Clinicopathological Presentation

We examined within-sample diversity (alpha diversity) from all cases positive for microbial presence as described in Rodriguez et al. [17]. In paired Student’s *t*-test analyses comparing tumor to its adjacent normal, within-sample diversity indices were not significantly different for any of the three cohorts (Figure 2).

We performed Student’s *t*-test for continuous variables and Chi-square test for categorical variables to examine differences in population characteristics and clinicopathological presentation in order to ascertain differences in microbial abundance and diversity profiles. Across cohorts, stratifying by sample type, there were no significant differences in the proportion of male-to-females, the age at diagnosis, or vital status. Microbial profile metrics such as absolute abundance (S), Shannon-Wiener diversity index (H), and evenness (H/ls(S)) measurements showed similar patterns in tumor and adjacent tissue samples among different racial groups across cohorts (Figure 3).

In HNSC, we observed small trends in absolute abundance or the number of species (*p* = 0.05) when comparing across groups. However, when stratifying by race, stage, vital status, morphology or tumor grade, the number of species did not differ significantly. Among HNSC patients, there were no significant differences, when considering relative abundance versus absolute abundance in microbial profiles at any taxonomic level. For two Actinomyces species, *Actinomyces pacaensis* (absolute abundance but not its relative abundance), and *Actinomyces myeri* (relative abundance but not its absolute abundance) were significantly different across the racial groups (*p* = 0.03). These differences remained after adjusting for cigarette exposure (cigarettes per day), and disappeared when adjusting for smoke (smoke years) (Table 3). 

There were no differences in clinicopathological presentation, including primary diagnosis and resection site, except for tumor stage when stratifying by race and sex observed in LIHC. LIHC microbial profiles were significantly different when stratifying by race and sex (Table 3). Species absolute abundance did not differ significantly, while within-sample diversity measures were significantly different among racial groups (Student’s *t*-test, Shannon Index of diversity, *p* = 0.01; evenness, *p* < 0.001). Asian patients within the LIHC cohort were significantly younger (mean 53.3 (10.5) years) compared to White (mean 65.8 (15.2) years) and the non-reported mix race group (mean 62.4 (11.9) years) counterparts (Wilcoxon, *p* < 0.001). Viral presence of HBV differed both by sex and race while the presence of HHV-4 did not. HBV presence was significantly higher among Asian patients (absolute and relative abundance *p* < 0.001). In pairwise tests, diversity index was observed to be significantly different between White and Asian patients (BH adjusted *p* = 0.009). We conclude that differences in bacterial diversity for LIHC differ by race, and this relationship is dependent on HHV-4 infection status (F(1, 53) = 4.8 *p* = 0.03). It must be noted that race proportions for LIHC were itself significantly different with no females of Asian background present.

Significant within-sample bacterial diversity was observed in STAD cohort where Black or African American patients fell within the highest diversity quartiles in both tumor and adjacent normal. There were significant differences with histopathological grade and site of resection, whereas there was no difference in tumor staging within the population. When stratifying by race, significant differences in primary diagnosis (*p* = 0.022) and age at diagnosis (*p* = 0.027), and site of resection (*p* = 0.022) were observed. EBV positive status was significantly higher among White females compared to other racial groups (13% non-White). Microbial species detected differed by sample type and race. Several species including, *Fusobacterium nucleatum*, *Lactobacillus amylovorrus*, *Lactobacillus salivarius*, *Campylobacter concisus*, *Lactobacillus fermentum*, and *Neisseria enlongata* were unique to tumor samples. *Cutibacterium acnes*, *Mycoplasma mycoides*, and *Ralstonia pickettii* presence were significantly different by race in tumor and Arthrobacter presence in adjacent normal.

### 2.5. Microbial Within-Sample Diversity Is Associated with Overall Survival

Based on the different bacterial compositions observed, we explored association between different variables, including within-sample diversity with the overall survival in HNSC, LIHC, and STAD. Among HNSC patients, there were no significant interactions between clinicopathological features and other variables, including sex and microbial diversity in relation to overall survival. An interaction was observed between sex and microbial diversity of the adjacent tissue when stratifying by vital status (Two-way ANOVA, F(2, 43) = 6.28, *p* = 0.004).

We then tested the relationship of diversity and overall survival stratifying by quartiles of Shannon Diversity index. Deceased HNSC male patients with high bacterial diversity (2.4 to 4.1) in the adjacent normal tissue were observed to have poorer survival compared to males with low diversity (0 to 1.44) while female counterparts had opposite effects. The differences in diversity and their relationships to overall survival are indicative of the duality of microbial presence, their interactions with the host, and with other microbes or environmental co-factors.

In LIHC, the relationship of diversity and overall survival stratifying by quartiles of Shannon Diversity index (low diversity = 0 to 0.4, intermediate-low = 0.4 to 1.4, intermediate-high = 1.4 to 2.3, and high diversity = 2.3 to 3.2) was explored (Figure 4). Intermediate diversity quartiles were associated with reduced overall survival compared to higher diversity quartiles (Kaplan–Meier, *p* = 0.033). In Cox logistic regression model, intermediate-low bacterial diversity was associated with more than double the risk compared to those at low (intermediate-low versus low diversity HR: 2.4, 95% CI: 1.2–5.0, *p* = 0.02). Analysis of deviance revealed significant interaction between age at diagnosis and HBV infection status with overall survival. Sex, race, and age were also associated with overall survival in LIHC. Among LIHC living males (n = 16), lower tumor bacterial diversity was associated with shorter days survived after diagnosis with opposite effect in females (n = 14) at the same diversity range. Contrary to HNSC, bacterial diversity in the adjacent tissue had a negative impact on females and favorable effect among males.

Overall survival in STAD was associated with bacterial diversity quartiles (Figure 5; low diversity = 0 to 0.8, intermediate-low = 0.8 to 1.6, intermediate-high = 1.6 to 2.4, and high diversity = 2.4 to 3.6), where higher diversity resulted in poorer survival outcomes (low versus high diversity HR: 2.8, 95% CI: 1.3–6.0, *p* = 0.01). Significant interaction between sex and tumor diversity was observed among those alive at time of censoring (F(2, 45) = 3.6, *p* = 0.03). Post hoc analyses revealed significant difference between males with high diversity compared to males with intermediate diversity indices (diff = −888, −1762 to −13.4, adjusted *p* value (BH) = 0.04). Overall survival was associated with demographic characteristics including race (*p* = 0.006) and age at diagnosis (*p* = 0.001). There was no correlation between overall survival and sex alone in the STAD cohort. We then tested the correlation between diversity, race, and age at diagnosis. We observed there was correlation between diversity index and race (Chi-sq = 27.4, df = 9, *p* = 0.001) and no correlation with age at diagnosis (rho = −0.008, 95% CI = −0.235, 0.063, *p* = 0.25). Analyses of variance were carried out to compare the relationship between survival days and diversity controlling for both age and race. We concluded that the relationship between diversity and survival days is dependent on the interaction with race, where Asian patients with lower diversity have better survival outcomes compared to White (HR = 0.16, 95% CI = 0.37, 0.7, *p* = 0.014). Hazards by clinicopathological features were also examined. In STAD, tumor histopathological stage was associated with diversity. On average, diversity index in STAD increased with tumor stages. Whites classified at tumor stage III had, on average, higher diversity indices than other racial groups. Tumor stage was in turn associated to survival days (Chi-sq = 43.3, *p* < 0.0). Relationship between survival days and stage was dependent on diversity (F(27, 129) = 2.3, *p* < 0.001). Cox proportional hazards revealed that those classified at tumor stages II to III with lower diversity indices survived longer days after diagnosis (HR = 0.037, 95% CI = 0.17, 0.8, *p* = 0.01).

## 3. Discussion

In high throughput sequencing data while microbial cancer associations have gained interest in recent years, insufficient attention has been paid to the potential of addressing racial disparities. Previous studies have identified bacterial diversity as a modulator of treatment response where a highly diverse gut microbiota that includes beneficial bacteria exerts beneficial treatment outcomes; compared to a gut with a less diverse ensemble and high prevalence of pathogenic bacteria which can have the opposite effects [16]. Other studies have found that gut and oral microbiota diversity contributes to racial differences in cancer [15] and in healthy adults [9,19,20]. Studies have focused on tumor viral detection or bacterial metagenomics profiling of the gut and oral microbiota. In this study, we examined the relationship between bacterial relative abundances and diversity to overall survival in three infection-associated cancers of the stomach, liver, and head and neck to determine if we could identify bacterial differential patterns associated with race and overall survival. To do this, we utilized previously created microbial abundance and diversity profiles from a total of 470 paired tumor and adjacent normal samples corresponding to 235 cases. Across the three cohorts, majority, 74% (n = 174) were self-reported as non-Hispanic White, 10% were Asian, 8% were Black or African American, and 8% were of no reported racial background and considered to be a mixed-race group. From these, 40% (n = 93) were females, and 55% (n = 131) were deceased. We compared bacterial diversity associations to clinical features including basic demographics (age at diagnosis, sex, race, and ethnicity), tumor stage, tumor grade, site of resection and histopathology. Smoke exposure (in head and neck cancers) and viral infection status of EBV, HBV, and HPV were also examined. Similar to previous studies of gut microbiota, we find that relative abundances of certain taxa in the tumor microenvironment are associated with race. Across all three infection-associated cancers examined here, the relationship between microbial diversity and overall survival differs by tissue type and are dependent on the interactions with sex and race. Interactions with race and sex also differ by vital status. We believe that the interactions with sex are perhaps related to the lower number or absence of females within the three cohorts examined; these associations are stronger in TCGA gastric and liver cancers than in cancers of the head and neck. We found that microbial tumor and adjacent tissue diversity indices were, on average, lower among persons of Asian background compared to their White counterparts. African Americans, on the other hand, had similar within-sample diversity to Whites in HNSC, lower in LIHC and higher in STAD. In addition, bacterial within-sample diversity relationship to overall survival had an opposite effect in STAD compared to LIHC, while no differences were observed in HNSC. We note that there were significant differences in the proportion of racial minorities as previously noted by Zhang et al. [21], and the presence or absence of viral agents associated with cancer initiation and aggressiveness [12] which could introduce bias.

In STAD cohort, higher bacterial diversity was associated with poorer outcomes (HR = 2.77, 95% CI = 1.3–6, *p* = 0.01) with differences among the population groups. For example, Asian persons had a lower risk of death compared to White persons (HR = 0.16, 95% CI = 0.04–0.7, *p* = 0.01). Yet, among those who were deceased, the average bacterial diversity in the tumor was higher among the Asian group compared to other groups. Among those living at time of censoring, there was a divergent association with days survived after diagnosis. Here, higher tumor tissue within-sample diversity appears to be associated with longer days while higher diversity in the adjacent tissue is associated with shorter days. We noted that Asian patients had, on average, the lowest diversity indices in the adjacent tissue. This could explain the protective effect Asian persons had within our STAD subset compared to White counterparts. It must be noted that the majority White sample population in our subset originated from Eastern Europe. Black or African Americans had higher bacterial diversity in the adjacent normal, which similarly was associated with their poorer outcomes. The bimodal bacterial diversity is suggestive of microbial species overturn (dysbiosis) and colonization in disease progression. We also examined microbial (viral and bacterial) presence, relative abundance, and co-occurrence patterns. In STAD, presence of EBV was associated with poorer survival (HR: 2.23, 95% CI 1.26, 3.94, *p* = 0.006). EBV prevalence was higher among non-Hispanic Whites who had the poorer outcomes within the STAD subset.

In LIHC, lower microbial diversity was associated with poorer overall survival (HR: 2.57, 95% CI: 1.2, 5.5, *p* = 0.14) while high diversity appeared to have a favorable effect overall. Interestingly, we observed that Asian and African American patients who were deceased at the time of censoring had lower bacterial diversities and opposite outcomes, where Asian patients appear to do poorer while African American patients have longer survival days compared to White patients. Bacterial diversity in the adjacent normal appears to have no effect while diversity in the tumor does. We also found a divergent effect of bacterial absolute and relative abundance on race, sex, and tumor stage. We tested for the possibility of site submitter interaction and found that cohorts have specific signatures by race which are associated with, or dependent on, age and sex. *Escherichia coli* absolute and relative abundance in LIHC was associated with submitter site and race. In the linear regression model accounting for interacting terms, the submitter site was associated with race. In pairwise analyses using the Wilcoxon rank sum test, sites with similar racial recruitment were highly correlated. We interpreted this as the difference in recruitment for racial minorities across different cohorts where the effect of the site is modulated by race. We observed that, for some bacterial species, there was significant interaction between bacterial abundance and the originating institution. Previous studies have suggested that microbial differential patterns are dependent on the submitting institution rather than other features—including demographics of age, sex, and race [18]. We found that, in LIHC, bacterial signatures patterns differ by submitting institution with high correlation between sites with similar enrollment patterns. We discovered an interaction between the submitting institution, sample type, and race. Although the possible effect of contamination at the submitter cannot be ruled out, our paired analyses design should account for this.

Small sample representation of racial minorities may influence our ability to detect hidden or masked associations. Future studies with larger sample sizes should confirm our findings. In addition, our approach could be applied to explore other cancer types including lung and breast. Our study concludes that bacterial within-sample diversity correlates with survival in HNSC, LIHC, and STAD TCGA cancers and these associations are dependent on the interactions with sex, race, and overall survival by cancer type.

## 4. Materials and Methods

### 4.1. TCGA Exome Data and Bioinformatics Pipeline

Paired primary tumor and solid tissue normal raw exome sequences in BAM file format were downloaded from The Cancer Genome Atlas (TCGA) consortium for the three recognized infection-associated cancers (head and neck, liver, and stomach). Other cancers commonly attributed to infectious etiology were not included on the basis of low pair sample availability. Microbial relative abundances and bacterial diversity results were derived by interrogating each BAM file using the bioinformatics workflow based on PathoScope 2.0 [22] as described in Rodriguez et al. [17]. Briefly, the pipeline includes extracting unmapped-to-human reads from the human tumor sequence files using SAMtools and Picard. Selected reads were adapter trimmed and quality filtered. Additional filtering was completed by computational subtraction against human reference genome (hg38) and then aligned to known microbial genome (custom library).

Diversity measurements per cancer type and across cancers were calculated. Shannon-Wiener Diversity Index (alpha) was used to compare between tumor and its paired adjacent normal. Furthermore, microbial profiles of tumor and adjacent normal pairs were correlated with de-identified identified relevant clinical and survival data downloaded from Genomes and Phenotypes (dbGaP) data commons under project Project-14778. Comparisons were made by demographic (age, sex, race/ethnicity) and clinicopathological characteristics (stage, grade, subsite), exposures (alcohol, tobacco), and survival outcomes.

### 4.2. Statistical Analyses

Bacterial within-sample (alpha) diversity was measured by Shannon-Wiener diversity index. Differences in bacterial within-sample diversity between tumors and their adjacent normal samples were compared using pairwise *t*-test and analysis of variance (ANOVA). Chi-square tests were used for categorical data. Cox proportional hazards regression analyses were used to evaluate the associations between diversity and overall survival. Tukey’s HSD post hoc tests were also carried out. All analyses were carried out in R.

## Figures and Tables

**Figure 1 molecules-26-05659-f001:**
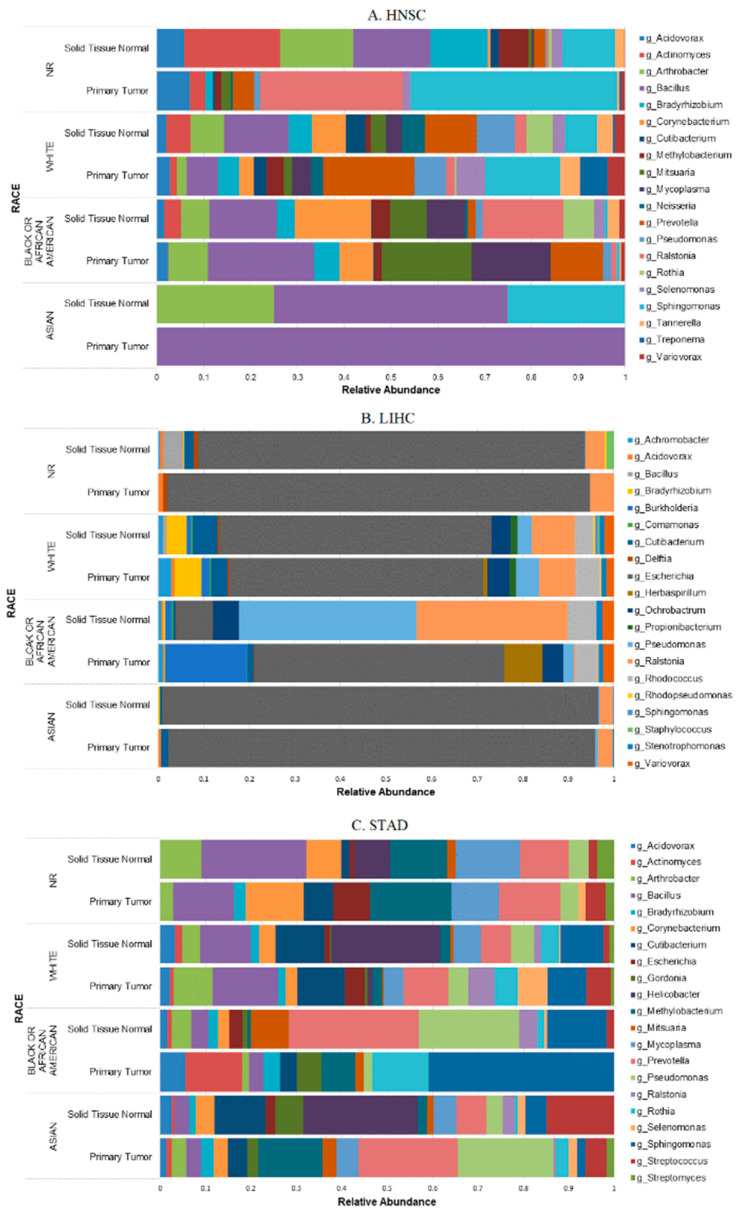
Microbial composition stratified by sample type and race at the genus level in HNSC (**A**), LIHC (**B**), and STAD (**C**) cancer cohorts. NR: not reported race.

**Figure 2 molecules-26-05659-f002:**
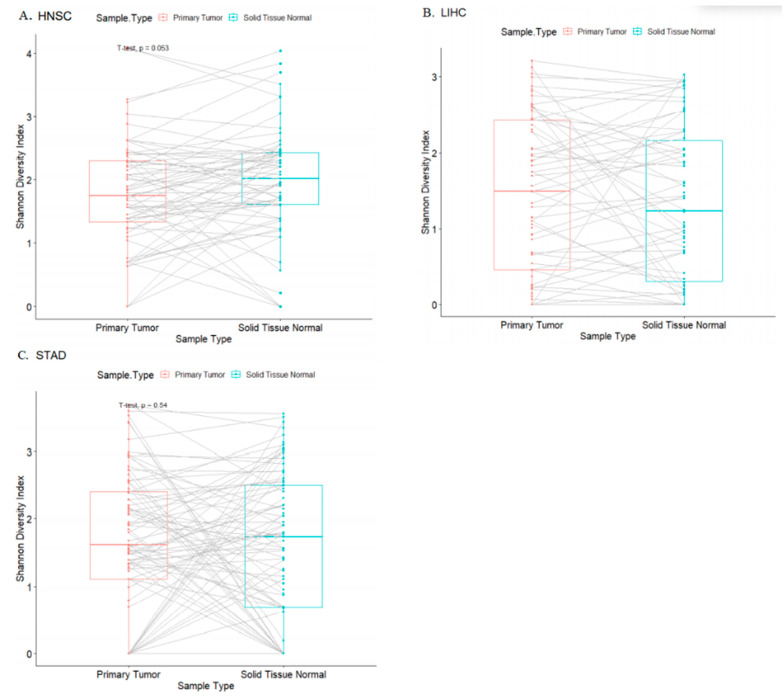
Bacterial within-sample diversity across cohorts comparing tumor to its paired adjacent normal tissue in HNSC (**A**), LIHC (**B**), and STAD (**C**) cancer cohorts.

**Figure 3 molecules-26-05659-f003:**
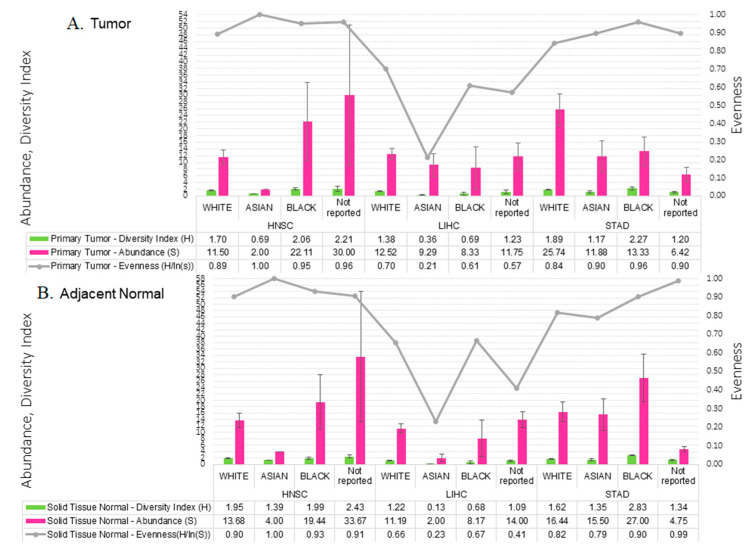
Microbial diversity profiles for infection associated cancers (HNSC, LICH, STAD) among different racial groups with standard error bars for tumor (**A**) and adjacent normal (**B**). Average absolute abundance (pink bars), within-sample diversity index as determined by Shannon-Wiener index (green bars), and evenness (y-axis) of species abundance per racial group in each cohort is shown as line graph. Evenness measure is being used as complement to Shannon-Wiener diversity index, where ‘1’ indicates complete evenness, and ‘0’ indicates high diversity.

**Figure 4 molecules-26-05659-f004:**
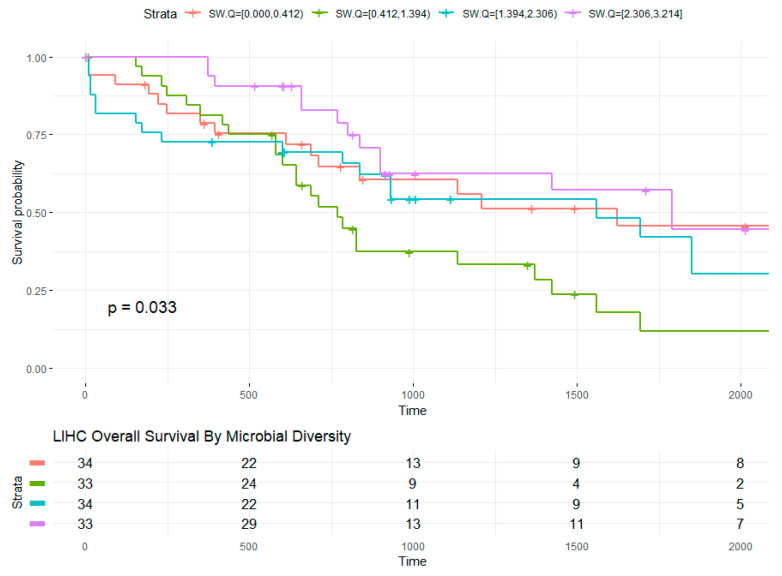
LIHC overall survival by microbial diversity. The survival curve based on Kaplan–Meier estimates (created with survminer) for the relationship between bacterial within-sample diversity and overall survival with global *p*-value is shown. Strata with risk set size is shown in table. Time in days survived after diagnosis or days to last follow up (censored). Censoring over time are delineated by “+” within the survival lines for each strata.

**Figure 5 molecules-26-05659-f005:**
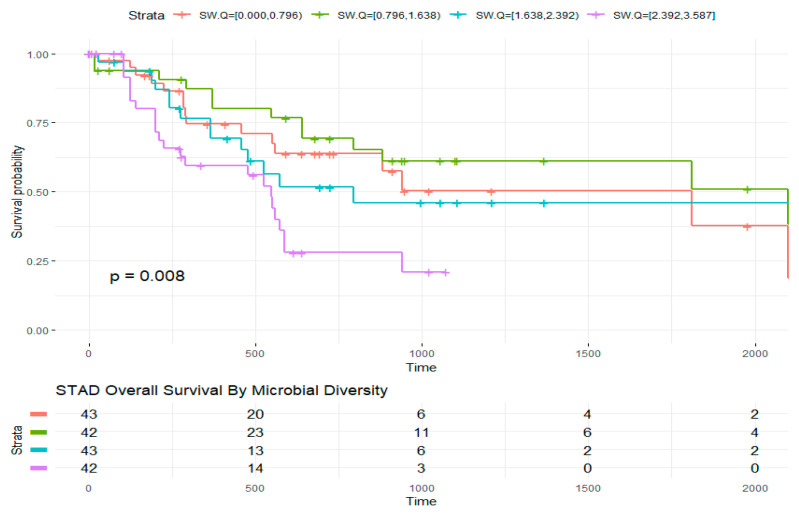
STAD overall survival by microbial diversity. The survival curve (created with survminer) for the relationship between within-sample bacterial diversity and overall survival with global *p*-value (*p* = 0.008) is shown. Strata with risk set size are shown in table. Time in days survived after diagnosis or days to last follow up (censored). Censoring over time are delineated by “+” within the survival lines for each strata.

**Table 1 molecules-26-05659-t001:** Population characteristics for HNSC, LIHC, and STAD cohorts.

	HNSC (n = 69)	LIHC (n = 81)	STAD (n = 85)
Sex			
Female	21 (30.4%)	35 (43.2%)	37 (43.5%)
Male	48 (69.6%)	46 (56.8%)	48 (56.5%)
Race			
Asian	1 (1.4%)	7(8.6%)	16 (18.8%)
Black	9 (13.0%)	6 (7.4%)	3 (3.5%)
Not reported	3 (4.3%)	4 (4.9%)	12 (14.1%)
White	56 (81.2%)	64 (79.0%)	54 (63.5%)
Age			
Mean (SD)	62.7 (12.2)	64.3 (14.7)	67.7 (10.5)
Missing	0 (0%)	2 (2.5%)	0 (0%)
Vital Status			
Deceased	49 (71.0%)	48 (59.3%)	34 (40.0%)
Living	20 (29.0%)	33 (40.7%)	51 (60.0%)

Note: Data are in n (%) unless specified otherwise. A proportion of samples had single or no bacteria. Cases with no identifiable bacteria in either sample were not considered. Vital status is the survival status of the patient.

**Table 2 molecules-26-05659-t002:** Microbial diversity profiles among infection associated cancers cohorts stratified by sample type.

	HNSC		LIHC		STAD	
	Primary Tumor (n = 69)	Adjacent Normal (n = 69)	Primary Tumor (n = 81)	Adjacent Normal (n = 81)	Primary Tumor (n = 85)	Adjacent Normal (n = 85)
Shannon						
Mean (SD)	1.76 (0.86)	1.96 (0.87)	1.23 (1.09)	1.08 (1.04)	1.67 (1.06)	1.57 (1.12)
Richness						
Mean (SD)	13.6 (21.3)	15.2 (19.8)	11.9 (13.0)	10.3 (11.2)	20.0 (30.2)	15.0 (20.3)
Evenness						
Mean (SD)	0.91 (0.13)	0.91 (0.13)	0.65 (0.31)	0.62 (0.28)	0.86 (0.18)	0.84 (0.22)
Median [Min, Max]	0.97 [0.41, 1]	0.96 [0.31, 1]	0.76 [0.07, 1]	0.71 [0.08, 1]	0.93 [0.33, 1]	0.96 [0.05, 1]
Missing	4 (5.8%)	4 (5.8%)	19 (23.5%)	24 (29.6%)	16 (18.8%)	18 (21.2%)
HBV Status						
Absent	69 (100%)	69 (100%)	75 (92.6%)	75 (92.6%)	85 (100%)	85 (100%)
Present	0 (0%)	0 (0%)	6 (7.4%)	6 (7.4%)	0 (0%)	0 (0%)
EBV Status						
Absent	63 (91.3%)	66 (95.7%)	80 (98.8%)	80 (98.8%)	60 (70.6%)	59 (69.4%)
Present	6 (8.7%)	3 (4.3%)	1 (1.2%)	1 (1.2%)	25 (29.4%)	26 (30.6%)
HPV Status						
Absent	63 (91.3%)	67 (97.1%)	81 (100%)	81 (100%)	85 (100%)	85 (100%)
Present	6 (8.7%)	2 (2.9%)	0 (0%)	0 (0%)	0 (0%)	0 (0%)

Note: Data are in n (%) unless specified otherwise. A proportion of samples had single or no bacteria. Cases with no identifiable bacteria in either sample were not considered.

**Table 3 molecules-26-05659-t003:** Shannon-Wiener diversity index in tumor and adjacent normal paired samples.

	Primary Tumor		Solid Tissue Normal		
	N = 235		N = 235		
Cohort	Female	Male	Female	Male	
	N = 93	N = 142	N = 93	N = 142	*p-*Value
HNSC					
White	1.75 (0.79)	1.68 (0.75)	2.05 (0.95)	1.90 (0.82)	
Asian	--	0.69 (0.00)	--	1.39 (0.00)	
Black	1.36 (0.03)	2.26 (1.05)	1.78 (0.17)	2.05 (1.08)	
Not reported	0.69 (0.00)	2.97 (1.12)	1.35 (0.00)	2.96 (1.08)	0.360
LIHC					
White	1.28 (1.01)	1.47 (1.10)	1.31 (1.02)	1.13 (1.05)	
Asian	--	0.36 (0.48)	--	0.13 (0.26)	
Black	0.05 (0.05)	1.01 (1.24)	0.00 (0.00)	1.01 (1.07)	
Not reported	2.02 (1.10)	0.43 (0.43)	0.92 (0.67)	1.26 (0.52)	0.001
STAD					
White	2.06 (0.87)	1.71 (0.97)	1.51 (1.09)	1.73 (1.05)	
Asian	1.28 (1.29)	1.08 (1.33)	1.56 (1.45)	1.19 (1.42)	
Black	--	2.27 (0.68)	--	2.83 (0.25)	
Not reported	1.35 (0.21)	1.15 (0.87)	1.19 (0.85)	1.39 (0.76)	0.007

Note: Table shows within-sample diversity index in HNSC, LIHC, and STAD by different racial groups stratified by sample type and sex. Females of Asian background are underrepresented in HNSC and LIHC and Black or African American females are underrepresented in STAD. Analyses of variance—after adjusting for sample type, race, and sex—show significant differences in alpha (within-sample) diversity among LIHC and STAD populations.

## Data Availability

Microbial profiles were derived from TCGA data package phs000178 versions v9.p8 and v10.p8. Results from this project are maintained by the University of Hawaii Bioinformatics Core, QHS John A. Burns School of Medicine. Microbial profile data are available upon request. All data were handled in accordance with signed data use agreement under data commons project no. 14778.

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
