# Peer review of "Bacterial Diversity Correlates with Overall Survival in Cancers of the Head and Neck, Liver, and Stomach"

_molecules, 2021, doi:10.3390/molecules26185659_

Round 1
Reviewer 1 Report
The paper is interesting idea based on TCGA big database. I have some questions for the authors.
- Why use LIHC as abbreviation of hepatocellular carcinoma?
- The sample of Asian and black population to condclude the difference of bacterial diversity and racial groups, especially in HNSC and LIHC.
- In table 1, what is the meaning of "deceased" or "living" viral status?
- How to confirm the EBV status in TCGA data? In addition, the percentage of EBV positive and HPV positive in HNSC is equal. Was it a common distribution in literature review? How many patients with co-infected EBV and HPV?
- Because differnet bacterial diversity between normal tissue and tumor tissue is correalted with survival, the authors can try to explain that why male HNSC with high bacterial diversity in noramal tissue have poorer survival?
- In page 9, why authors commented "HPV status appeared to have an effect on ......and smoking status". Please clarify it clearly.
- The exposure of antibioitics within weeks before sample approach imapacts on bacterial diversity in tumor and normal tissue. Was available records of antibioitcs inTCGA, such as treatment fo H. Pylori infection?
- What is the percentage of HCV infection in LIHC?
Reviewer 2 Report
Very interesting paper where to be reviews the role of infectious agents against the appearance of certain cancers: liver, head and neck, gastric.
The authors evaluated various microbial sequences from a set of cancers associated with infectious agents.
In my opinion the methodology used is adequate to the objectives of the study as well as the analysis of the obtained data.The results well presented and the discussion and conclusions , correct.
The article can be published in the current version.
Author Response
We thank the Reviewer for carefully reading our manuscript and supporting our research work!